# Towards Effective Federated Graph Anomaly Detection via Self-boosted Knowledge Distillation

Jinyu Cai*
National University of Singapore
Singapore
jinyucai@nus.edu.sg

Yunhe Zhang*†
University of Macau
Macau, China
zhangyhannie@gmail.com

Zhoumin Lu
Northwest Polytechnical University
Xi'an, China
walker.zhoumin.lu@gmail.com

Wenzhong Guo
Fuzhou University
Fuzhou, China
guowenzhong@fzu.edu.cn

See-Kiong Ng
National University of Singapore
Singapore
seekiong@nus.edu.sg

## Abstract

Graph anomaly detection (GAD) aims to identify anomalous graphs that significantly deviate from other ones, which has raised growing attention due to the broad existence and complexity of graph-structured data in many real-world scenarios. However, existing GAD methods usually execute with centralized training, which may lead to privacy leakage risk in some sensitive cases, thereby impeding collaboration among organizations seeking to collectively develop robust GAD models. Although federated learning offers a promising solution, the prevalent non-IID problems and high communication costs present significant challenges, particularly pronounced in collaborations with graph data distributed among different participants. To tackle these challenges, we propose an effective federated graph anomaly detection framework (FGAD). We first introduce an anomaly generator to perturb the normal graphs to be anomalous and train a powerful anomaly detector by distinguishing generated anomalous graphs from normal ones. We subsequently leverage a student model to distill knowledge from the trained anomaly detector (teacher model), which aims to maintain the personality of local models and alleviate the adverse impact of non-IID problems. Additionally, we design an effective collaborative learning mechanism that facilitates the personalization preservation of local models and significantly reduces communication costs among clients. Empirical results of diverse GAD tasks demonstrate the superiority and efficiency of FGAD.

## CCS Concepts

• **Computing methodologies → Artificial intelligence**; **Neural networks**; • **Security and privacy → Intrusion/anomaly detection and malware mitigation**;

---

*Both authors contributed equally to this research.
†Corresponding author.

---

## Keywords

Unsupervised Learning, Anomaly Detection, Federated Learning, Graph Neural Networks

**ACM Reference Format:**
Jinyu Cai, Yunhe Zhang, Zhoumin Lu, Wenzhong Guo, and See-Kiong Ng. 2024. Towards Effective Federated Graph Anomaly Detection via Self-boosted Knowledge Distillation. In *Proceedings of the 32nd ACM International Conference on Multimedia (MM '24), October 28-November 1, 2024, Melbourne, VIC, Australia.* ACM, New York, NY, USA, 10 pages. https://doi.org/10.1145/3664647.3681415

## 1 Introduction

Anomaly detection [5, 35] is a fundamental research problem in machine learning, which has been extensively explored in various domains, *e.g.*, images [2, 17] and time-series data [1, 6]. In the real world, graph-structured data [21–23, 34, 44] is commonly available due to its exceptional ability to represent complicated relationship information among entities [29, 30, 54]. This is particularly evident in domains like social networks and medical applications. In return, graph anomaly detection (GAD) [7, 32], which aims to identify graphs that exhibit significant deviations from other normal graphs, has raised broad attention in recent years. With the advancement of graph neural networks (GNNs) [4, 15, 47], GAD has made remarkable strides and demonstrated promising performance in detecting anomalies across many real-world scenarios with natural graph-structured data, *e.g.*, social networks and bioinformatics.

In realistic collaborative efforts among different companies and organizations, a common demand is an attempt to share their knowledge to detect anomalies more accurately. Although existing GAD approaches [11, 31, 39, 50] simplify coordination with centralized training, it introduces a critical privacy leakage risk as it typically requires all participants to provide their own data to train a global model, as shown in Figure 1(a). Graph data may encompass sensitive information that the participants are not willing to share, *e.g.*, private relationships in social networks, which will hinder collaborations. Therefore, an urgent imperative emerges to investigate approaches that facilitate collaboration between GAD models distributed to different participants while protecting their privacy.

As the emerging technique in machine learning, federated learning (FL), as shown in Figure 1(b), enables collaboration between different participants with the consideration of privacy-preserving. Clients in FL are only required to share their network parameters

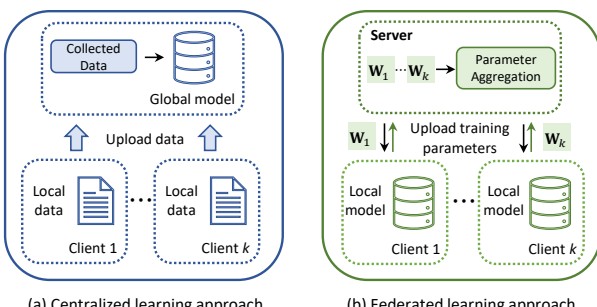

(a) Centralized learning approach      (b) Federated learning approach

**Figure 1: Overview of the centralized learning and federated learning frameworks.**

with the server rather than their local data, which will prevent the leakage of sensitive information in participants. Classical FL methods, such as FedAvg [33] and FedProx [20], have become the paradigm of collaborative learning across various domains [16, 45]. To facilitate collaborative training of GNN models for graph data across clients, federated graph learning (FGL) [10, 24, 51] has also been widely studied in recent years. FGL methods [38, 46] integrate GNNs with FL methods to collaboratively learn representations for complicated graph data distributed in various clients, and have demonstrated superiority in many downstream tasks, *e.g.*, graph classification. Hence, an intuitive approach to address the above issue is to integrate the existing advancements in FL and FGL with general anomaly detection techniques, *e.g.*, deep one-class classification (DeepSVDD) [37].

However, this solution may encounter the following challenges:

(1) The graph data distributed in various clients often exhibit significant heterogeneity and non-IID property [12, 46], *e.g.*, containing different graph structures or feature dimensions. These factors place a higher demand on maintaining the validity of the local models for their own data, *e.g.*, personalization.

(2) It is difficult to learn a universal hypersphere as the decision boundary for highly heterogeneous graph data under the federated learning setting. Besides, such non-IID graphs across clients hardly conform to the assumption in DeepSVDD that their latent distribution could follow a universal hypersphere.

(3) Existing collaborative learning mechanisms, *e.g.*, FedAvg [33], require transmitting all network parameters of each client in a single communication round, which brings substantial communication costs in applications.

These challenges naturally lead us to a research question: ***Can we design an FL-based collaborative GAD framework to detect anomalous graphs with non-IID properties effectively?***

In this paper, we propose an effective federated graph anomaly detection (FGAD) framework, as shown in Figure 2, to answer this research question. To improve the anomaly detection capability in the local model, we introduce an anomaly generator that perturbs normal graphs to be anomalous, and unsupervised train a classifier to identify anomalies from normal graphs. The generated anomalous graphs are encouraged to be diverse but resemble normal ones through iterations, so that more robust decision boundaries can be learned in a self-boosted manner. To alleviate the adverse impact

of non-IID problems, we propose to preserve the personalization of each client by leveraging knowledge distillation. Specifically, we introduce a student model to distill the knowledge from the trained teacher anomaly detector. The student model only takes the normal graphs as the input, with the aim of aligning its predicted distributions with that of the teacher model. Moreover, we further design an effective collaborative learning mechanism, in which the student and teacher models share the same backbone network to streamline the capacity of local models. Furthermore, we engage only the parameters of the student head rather than the entire model in collaborative learning, which allows the teacher model to preserve the personalization of a client. In this way, we not only alleviate the adverse impact of non-IID property, but also reduce the communication costs between clients and server during collaborative learning. The contributions of this paper are:

- We, for the first time, explore the GAD problem on non-IID graphs under federated learning setup, and propose an effective federated graph anomaly detection (FGAD) method.
- We introduce a self-boosted distillation module, which not only promotes the detecting capability by identifying self-generated anomalies, but also maintains the personalization of local models from knowledge distillation to alleviate non-IID problems.
- We propose an effective collaborative learning mechanism that streamlines the capacity of local models and reduces communication costs with the server.
- We establish a comprehensive set of baselines for federated graph anomaly detection. Extensive experiments also validate the effectiveness of our FGAD.

## 2 Related Works

### 2.1 Graph Anomaly Detection

Graph anomaly detection (GAD) [3, 26, 32, 42, 52] refers to detecting abnormal graphs that significantly differ from other normal ones, which have received growing attention in recent years owing to the ubiquitous prevalence of graph-structured data in real-world scenarios, *e.g.*, social networks [25, 28]. There are many works that advance the research on GAD. For instance, Zhao et al. [53] investigated graph-level anomaly detection issues by integrating graph isomorphism network (GIN) [47] with deep one-class classification (DeepSVDD) [37]. Qiu et al. [36] leveraged neural transformation learning to develop a more robust GAD model to overcome the performance flip issue. Ma et al. [31] utilized knowledge distillation to capture more comprehensive normal patterns from the global and local views for detecting graph anomalies.

Although these GAD methods have achieved remarkable success, they primarily rely on a centralized training paradigm. In real-world collaborative scenarios, the graph data is often distributed across various clients from data owners, which necessitates the transmission of local graph data to a central server during their practical collaborations. Unfortunately, this process can potentially expose sensitive information and pose severe privacy risks. Additionally, the inherent non-IID property in the graph data distributed across diverse clients presents yet another formidable challenge. As a result, effective solutions to address these challenges remain an open research problem.

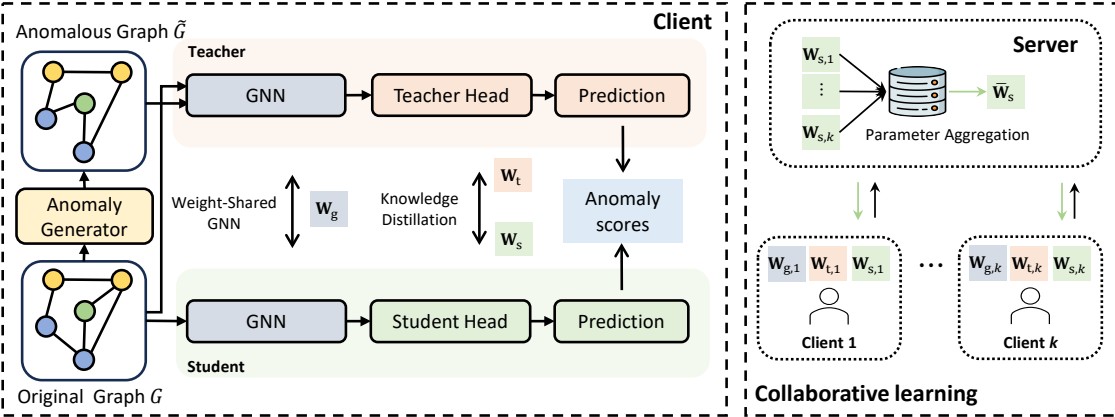

**Figure 2: Overview of the FGAD framework. Note that the teacher model utilizes both normal and generated anomalous graphs for training an anomaly detector, while the student model only inputs normal graphs for the distillation of normal patterns.**

## 2.2 Federated Graph Learning

Federated learning (FL) approaches [12, 19, 49], such as FedAvg [33], FedProx [20], provide a promising solution for collaboratively training models with data distributed in different clients, while preserving their privacy. In FL, clients only share their network parameters rather than data with the central server, which mitigates the privacy leakage risk and enables clients to share and leverage knowledge from others. As an emerging technique, FL has not only made remarkable advancements in handling image [8, 18, 48] and time series data [27, 41], but also raised attention to graph data [43, 51], where collaborative efforts are significantly more challenging due to the complex structural information and heterogeneous characteristic of graphs compared to other data types.

Federated graph learning (FGL) [51] aims to facilitate the collaboration of GNNs distributed in multiple remote clients to meet the requirement of handling complicated non-IID graph data that widely exist in many real-world scenarios, *e.g.*, social networks, medical, and biological data. For example, Xie et al. [46] studied the federated learning issue on non-IID graphs by integrating clustered federated learning with GIN, which achieves effective collaborations for distributed GINs. Tan et al. [38] designed a structural knowledge-sharing mechanism to facilitate the federated graph learning process. Although existing FGL methods have been validated for many tasks, such as graph classification, their effectiveness in addressing the intricate unsupervised graph anomaly detection remains an ongoing area to be explored. While it is possible to extend these FL/FGL [20, 33, 38, 46] methods to address GAD tasks by integrating them with classical anomaly detection methods like DeepSVDD [32, 37], it is imperative to acknowledge some significant challenges, *e.g.*, the adverse impact of the non-IID problem across different clients and the communication costs of transmitting complex GNN model parameters during collaborative learning.

## 3 Methodology

### 3.1 Preliminary and Problem Formulation

**Notation:** Let $D=\{G_1, \ldots, G_N\}$ denote a graph dataset which consists of $N$ graphs, and each graph $G_i=\{V_i, E_i\}$ in the graph set

comprises a node set $V_i$ and edge set $E_i$. Typically, assume the number of nodes in a graph $G_i$ is $n_i=|V_i|$, an adjacency matrix $\mathbf{A}_i \in \{0, 1\}^{n_i \times n_i}$ is used to represent the topology of graph $G_i$. Besides, let $\mathbf{x}_v \in \mathbb{R}^d$ denote the attribute vector for node $v \in V_i$, $\mathbf{X}_i \in \mathbb{R}^{n_i \times d}$ is used to represent the attribute matrix of graph $G_i$.

*Graph Neural Networks:* Graph neural networks (GNNs), which iteratively learn representations with neighborhood aggregation and message propagation, is a widely used paradigm of learning representation for graph-structured data in many downstream tasks. In this paper, we leverage the graph isomorphism network (GIN) [47], a widely used GNN backbone, to learn graph representation for anomaly detection tasks. Generally, in each layer of a GIN, the node representation is updated by aggregating its neighborhood information. For instance, in the $k$-th layer of GIN, the learned aggregated features $\mathbf{a}_v^{(k)}$ for node $v$ can be formulated as:

$$\mathbf{a}_v^{(k)} = \text{AGGREGATE}(\{\mathbf{h}^{(k-1)}(u), u \in \tilde{\mathcal{N}}(v)\}), \quad (1)$$

where $\text{AGGREGATE}(\cdot)$ indicates the aggregation function, and $\tilde{\mathcal{N}}(v)$ represents the neighbor node set of node $v$. Then, the node feature $\mathbf{h}_v^{(k)}$ in the $k$-th layer is obtained by combing the node feature learned in the $(k-1)$-th layer with the aggregated feature:

$$\mathbf{h}_v^{(k)} = \sigma(\text{COMBINE}(\mathbf{h}_v^{(k-1)}, \mathbf{a}_v^{(k)})), \quad (2)$$

where $\sigma(\cdot)$ denotes the activation function, *e.g.*, LeakyReLU. Particularly, the initial feature $\mathbf{h}_v^{(0)}$ for node $v$ is set as $\mathbf{h}_v^{(0)}=\mathbf{x}_v$. Consequently, we can obtain the representation for a graph $G$ based on the learned features of all nodes within $G$ as follows:

$$\mathbf{h}_G = \mathcal{R}(\text{CONCAT}(\mathbf{h}_v^{(k)}, k \in \{1, \ldots, K\}), v \in G), \quad (3)$$

where $K$ is the number of GIN layers, and $\text{CONCAT}(\cdot)$ denotes the concatenate operation that stacks the graph representation learned across all $K$ layers. $\mathcal{R}(\cdot)$ denotes the readout function that obtains the graph-level representation by aggregating the node features within a graph, and we choose sum-readout in this paper. Note that for convenience, we use $\text{GIN}(\cdot)$ to simply represent a GIN model containing the above three operations in the following sections.

**Problem Formulation:** The objective of the GAD under the FL setup is to facilitate collaboration among clients, which allows each participant to enhance their GAD models by leveraging knowledge from others without exposing private data. Given $C$ clients, the collective graph dataset is denoted as $D=\{D_c\}_{c=1}^C$, where each client possesses its own graph set $D_c$. A prevalent paradigm in GAD [32] is that all graphs within the client, *i.e.*, $\forall G_i \in D_c$, are deemed as "normal". The model is trained to capture this normality so that the trained model can identify when an "anomalous" graph $\tilde{G}$ deviates significantly from the distribution of $D_c$ by some pre-defined assumptions, *e.g.*, the hypersphere decision boundary in DeepSVDD [37]. On the contrary, in this paper, we attempt to develop an anomaly detector that can adaptively learn decision boundaries rather than relying on the strong assumption of the shape of the latent distribution. This can be regarded as solving:

$$\underset{\mathbf{w}^{(1)},...,\mathbf{w}^{(C)}}{\text{minimize}} \quad \frac{1}{C} \sum_{c=1}^{C} \frac{|D_c|}{|D|} (\ell_c(y, f_{\mathbf{w}^{(c)}}(G)) + \ell_c(\tilde{y}, f_{\mathbf{w}^{(c)}}(\tilde{G}))), \quad (4)$$

where $|D|$ and $|D_c|$ denote the total number of graphs and that of in $c$-th client. $\{G, y\}$ represents the normal graph labeled with $y=1$, and $\{\tilde{G}, \tilde{y}\}$ represents the anomalous graph labeled with $y=0$. $\ell_c(\cdot)$ denotes the local loss function of $c$-th client, *e.g.*, binary cross-entropy loss. $f_{\mathbf{w}^{(c)}}(\cdot)$ is the GIN-based neural network of $c$-th client, which is parameterized by $\mathbf{w}^{(c)}$. However, tackling this problem presents the following challenges:

1) GAD is generally an unsupervised task in which only normal graphs (w/o labels) are accessible. How will we produce high-quality anomalous graphs for training local anomaly detectors?
2) In the context of FL-based GAD, how can we alleviate the adverse impact of the prevalent non-IID problem across clients?
3) Transmitting all network parameters following conventional FL methods may limit the scalability given the complexity of GIN. Therefore, how do we reduce communication costs in collaborative learning while maintaining the validity of local models?

## 3.2 Self-boosted Graph Knowledge Distillation

The first challenge here is to produce anomalous graphs without using any supervised information. To resolve this, we propose a graph anomaly generator, denoted as $\mathcal{G}_{\mathbf{w}_a}(\cdot)$, to generate anomalous graphs by perturbing the graph structure of normal graph $G$. For each client, we aim to generate an anomalous graph set $\tilde{D}_c=\{\mathbf{X}_c, \tilde{\mathbf{A}}_c\}$ in an unsupervised manner by feeding it with normal graph set $D_c$[1]. To ensure diversity in the generated anomalous graphs, we utilize variational graph auto-encoder (VGAE) [14] to build the anomaly generator. Specifically, we first learn a latent Gaussian distribution $\mathcal{N}(\boldsymbol{\mu}_c, \boldsymbol{\sigma}_c^2)$, which can be determined as:

$$\boldsymbol{\mu}_c = \text{GIN}_{\mu}(\mathbf{X}_c, \mathbf{A}_c), \quad \log \boldsymbol{\sigma}_c = \text{GIN}_{\sigma}(\mathbf{X}_c, \mathbf{A}_c), \quad (5)$$

where $\text{GIN}_{\mu}(\cdot)$ and $\text{GIN}_{\sigma}(\cdot)$ denote two distinct GINs in anomaly generator, and $\boldsymbol{\mu}_c$ and $\boldsymbol{\sigma}_c$ explicitly parameterize the following inference model:

$$q(\tilde{\mathbf{Z}}_c|\mathbf{X}_c, \mathbf{A}_c) = \prod_{i=1}^{|D_c|} q(\mathbf{Z}_c^{(i)}|\mathbf{X}_c, \mathbf{A}_c), \quad (6)$$

[1]Note that we slightly abuse notations $\mathbf{X}_c$, $\tilde{\mathbf{A}}_c$, and $\mathbf{A}_c$ to represent the relevant attribute and adjacency matrices in the $c$-th client.

where $q(\tilde{\mathbf{Z}}_c^{(i)}|\mathbf{X}_c, \mathbf{A}_c)=\mathcal{N}(\tilde{\mathbf{Z}}_c^{(i)}|\boldsymbol{\mu}_c^{(i)}, \text{diag}(\boldsymbol{\sigma}_c^{(i)2}))$, and it allows us to sample from a wide range in the latent space thereby facilitating the diverse anomalous graph generation. Here, we employ the reparametrization trick [13] to address the obstacle of gradient propagation in the sample operation. Consequently, the generated adjacency matrix can be calculated by:

$$\tilde{\mathbf{A}}_c = \mathcal{T}(\tilde{\mathbf{Z}}_c^{\top}\tilde{\mathbf{Z}}_c), \quad \tilde{\mathbf{Z}}_c = \boldsymbol{\mu}_c + \epsilon \exp(\boldsymbol{\sigma}_c), \quad \epsilon \sim \mathcal{N}(\mathbf{0}, \mathbf{1}), \quad (7)$$

where $\mathcal{T} : \mathbb{R} \rightarrow [0, 1]$ represents the element-wise transformation operations such as Sigmoid($\cdot$), and $\epsilon$ represents a random Gaussian noise that follows the standard normal distribution $\mathcal{N}(\mathbf{0}, \mathbf{1})$.

Intuitively, allowing the generated graphs to closely resemble normal graphs while remaining as anomalies is beneficial in training a robust and powerful anomaly detector, as it forces the model to distinguish those subtle deviations from the normal patterns. Therefore, we propose to optimize the anomaly generator by minimizing the following objective:

$$\ell_g^c(\mathbf{A}_c, \tilde{\mathbf{A}}_c) = - \sum_{i,j} (\mathbf{A}_c^{ij} \log(\tilde{\mathbf{A}}_c^{ij}) + (1 - \mathbf{A}_c^{ij}) \log(1 - \tilde{\mathbf{A}}_c^{ij})), \quad (8)$$

where $\mathbf{A}_c^{ij}$ denotes the presence (1) or absence (0) of an edge between nodes $i$ and $j$, $\ell_g^c$ is the binary-cross entropy loss function. Subsequently, we can train an anomaly detector with the normal and generated anomalous graph sets for the local client as follows:

$$\ell_{\text{ad}}^c = l_{\text{ce}}(y_c, \text{Proj}(f_{\mathbf{w}_g}(\mathbf{X}_c, \mathbf{A}_c))) + l_{\text{ce}}(\tilde{y}_c, \text{Proj}(f_{\mathbf{w}_g}(\mathbf{X}_c, \tilde{\mathbf{A}}_c))), \quad (9)$$

where $\tilde{\mathbf{A}}_c=\mathcal{G}_{\mathbf{w}_a}(\mathbf{X}_c, \mathbf{A}_c)$, $l_{\text{ce}}(\cdot)$ is the cross-entropy loss, and $f_{\mathbf{w}_g}(\cdot)$ denotes the GIN backbone that learns graph representation by feeding with graph data. $\text{Proj}(\cdot)$ is the MLP-based projection head that maps the graph representation learned from $f_{\mathbf{w}_g}(\cdot)$ into the predicted logits. Notably, we simply set pseudo labels for the normal graphs and the generated anomalous graphs, *i.e.*, $y_c=1$ and $\tilde{y}_c=0$.

Hence, we can train an anomaly detector in an unsupervised manner by minimizing the following objective function:

$$\ell_{\text{pt}} = \frac{1}{C} \sum_{c=1}^{C} \frac{|D_c|}{|D|} (\ell_{\text{ad}}^c + \ell_g^c), \quad (10)$$

where $\ell_g^c$ attempts to generate anomalous graphs that closely resemble normal ones, while $\ell_{\text{ad}}^c$ aims to identify those generated anomalous graphs. Therefore, we produce diverse anomalous graphs to learn a powerful anomaly detector in such a self-boosted style, and the two objectives mutually improve each other during training.

However, in the context of federated learning, the graph data across different clients are often heterogeneous and exhibit non-IID property. Such characteristics can potentially affect the anomaly detection performance of local models, *i.e.*, the second challenge. To alleviate the adverse impact of the non-IID problem, we propose a graph knowledge distillation framework, which is designed to preserve the personalization of the local model during collaborative learning. Specifically, we regard the previously pre-trained anomaly detector as the "teacher" model, and introduce a "student" model that aims to distill the knowledge from the teacher model and achieve collaboration between clients.

The network architecture of the student model is similar to the teacher model, which consists of a GIN backbone and a projection head. Since the purpose of the student model is to mimic the predictions of the teacher model for normal data, only normal graphs

are considered in the knowledge distillation. The predicted logits of the teacher and student models are computed as follows:

$$Q_{c,\text{t}} = \text{Proj}_{\text{t}|\mathbf{w}_\text{t}}(f_{\mathbf{w}_g}(\mathbf{X}_c, \mathbf{A}_c)), \quad Q_{c,\text{s}} = \text{Proj}_{\text{s}|\mathbf{w}_\text{s}}(f_{\mathbf{w}_{g'}}(\mathbf{X}_c, \mathbf{A}_c)), \quad (11)$$

where $f_{\mathbf{w}_g}(\cdot)$, $\text{Proj}_{\text{t}|\mathbf{w}_\text{t}}(\cdot)$ and $f_{\mathbf{w}_{g'}}(\cdot)$, $\text{Proj}_{\text{s}|\mathbf{w}_\text{s}}(\cdot)$ are the backbone networks and projection heads of teacher and student models respectively. Note that $\text{Proj}_{\text{t}|\mathbf{w}_\text{t}}(\cdot)$ is actually the same as the projection head $\text{Proj}(\cdot)$ in Eq. (9). Subsequently, the student model distills the knowledge from the teacher model by matching its predicted logits with those of the teacher model, described as follows:

$$\ell_{\text{kd}}^c = \frac{1}{|D_c|} \sum_{i \in D_c} KL(\text{softmax}(\mathbf{Q}_{c,\text{t}}^{(i)}/\tau), \text{softmax}(\mathbf{Q}_{c,\text{s}}^{(i)}/\tau)), \quad (12)$$

where $KL(\cdot, \cdot)$ denotes the Kullback-Leibler divergence, which is applied to measure the discrepancy between the distribution of the predicted logits from teacher and student models. $\text{softmax}(\cdot)$ is the softmax function, *i.e.*, $\text{softmax}(q_i/\tau) = \frac{\exp(q_i/\tau)}{\sum_j \exp(q_j/\tau)}$, and $\tau$ is the temperature factor that controls the smoothness of the distillation.

## 3.3 Parameter-efficient Collaborative Learning

Based on the design of the self-boosted graph knowledge distillation module, the objective function of all clients is defined as follows:

$$\mathcal{L}_{\text{total}} = \frac{1}{C} \sum_{c=1}^{C} \frac{|D_c|}{|D|} (\ell_{\text{ad}}^c + \lambda \ell_{\text{g}}^c + \gamma \ell_{\text{kd}}^c), \quad (13)$$

where $\lambda$ and $\gamma$ are the two trade-off parameters. In federated learning, let $\mathbf{W}^{(c)} = \{\mathbf{w}_a^{(c)}, \mathbf{w}_g^{(c)}, \mathbf{w}_{g'}^{(c)}, \mathbf{w}_\text{t}^{(c)}, \mathbf{w}_\text{s}^{(c)}\}$ denote the parameter set of the $c$-th client, the conventional solution achieves collaboration by uploading the network parameters to the server and subsequently distribute the aggregated network parameters to each client. However, there are several problems with this solution. First, the high parameter complexity of a GIN-based backbone can limit the scalability of the model during the parameter aggregation process. Second, the transmission of all network parameters may introduce non-IID problems, and affect the performance of local models trained on different graph data across clients.

To address these issues, we propose an effective collaborative learning mechanism, which is described in Figure 2. Specifically, We let the teacher and student models share the same GIN backbone for learning graph representation, *i.e.*,

$$\mathbf{Z}_c = f_{\mathbf{w}_g}(\mathbf{X}_c, \mathbf{A}_c) = f_{\mathbf{w}_{g'}}(\mathbf{X}_c, \mathbf{A}_c), \quad (14)$$

where $\mathbf{Z}_c$ denotes the learned graph representation that is shared as the input to the projection heads of teacher and student. This operation not only reduces the complexity of the local model, but also simplifies the knowledge distillation of the student model. We upload only the parameter set $\mathbf{w}_\text{s}^{(c)}$ of the student head for collaboration instead of uploading all the network parameters, *i.e.*, the parameter aggregation in the server is formalized as follows:

$$\bar{\mathbf{w}}_\text{s} = \sum_{c=1}^{C} \frac{|D_c|}{|D|} \mathbf{w}_\text{s}^{(c)}, \quad (15)$$

where $\bar{\mathbf{w}}_\text{s}$ denotes the aggregated parameters in the server. The proposed collaborative learning mechanism not only streamlines the capacity of local models, but also significantly reduces the communication costs, which addresses the third challenge. To facilitate

the understanding of the proposed FGAD method, we summarize its detailed training process in Algorithm 1. The collaboration between clients via the student model is performed in the following steps:

- Each client performs graph knowledge distillation independently, updating its network parameters, and uploads the network parameters of the student head to the server.
- The server then aggregates the network parameters following Eq. (15), and distributes the aggregated network parameters to each client.

To demonstrate the efficiency of FGAD, we further conduct theoretical and empirical analysis (refer to **Appendix 4-5**).

---

**Algorithm 1** Training procedure of FGAD.

---

**Input:** Graph set $D = \{D_c\}_{c=1}^C$, number of clients $C$, number of GNN layers $K$, learning rate $\alpha$, total epochs $\mathcal{T}$.
**Output:** The overall graph anomaly detection performance.
1: Initialize the parameter sets $\{\mathbf{W}^{(c)}\}_{c=1}^C$ for each local model;
2: Pretrain the local model in each client with Eq. (10);
3: **while** not converge **do**
4:    **for** $t = 1, 2, \ldots, \mathcal{T}$ **do**
5:       **for** $c = 1, \ldots, C$ **do**
6:          Generate anomalous graph set $\tilde{\mathbf{D}}$ with Eqs. (5), (6), (7);
7:          Compute loss items $\ell_{\text{ad}}^c$, $\ell_{\text{g}}^c$, $\ell_{\text{kd}}^c$ with Eqs. (8), (9), (12);
8:       **end for**
9:       Back-propagation and update each local model via minimizing Eq. (13);
10:    **end for**
11:    Upload the parameter sets $\{\mathbf{w}_\text{s}^{(c)}\}_{c=1}^C$ of student model in each client to the server;
12:    Compute aggregated network parameters $\bar{\mathbf{w}}_\text{s}$ with collaborative learning following Eq. (15);
13:    Distribute parameter set $\bar{\mathbf{w}}_\text{s}$ to the local model of each client;
14: **end while**
15: Evaluate the anomaly detection performance in each client and aggregate their results;
16: **return** The overall graph anomaly detection performance.

---

## 4 Experiment

## 4.1 Experimental Setup

***Datasets.*** We evaluate the performance of FL-based graph anomaly detection on non-IID graphs through two distinct experimental setups: (1) single-dataset and (2) multi-dataset scenarios.

- **Single-dataset:** We distribute a single dataset across multiple clients, each of which possesses a unique subset of the dataset. This setup allows us to assess the effectiveness when clients collaborate on a shared dataset. We employ three social network datasets, including IMDB-BINARY, COLLAB, and IMDB-MULTI, to conduct this experiment.
- **Multi-dataset:** We broaden our evaluation by considering various datasets distributed in multiple clients, and each of them holds a specific dataset. We consider social network data (SO-CIALNET) as well as molecular (MOLECULES), biochemical

**Table 1: Anomaly detection performance (mean(%) ± std(%)) under the single-dataset setting. Note that the best performance is marked in Bold, and the last column shows the number of transmitted parameters in collaborative learning.**

| Methods | IMDB-BINARY | | COLLAB | | IMDB-MULTI | | # Parameters |
|---|---|---|---|---|---|---|---|
| | AUC | AUPRC | AUC | AUPRC | AUC | AUPRC | |
| Self-train | 41.58±1.34 | 47.43±1.39 | 46.96±1.80 | 30.87±0.62 | 52.39±1.31 | 32.74±0.60 | N/A |
| FedAvg [33] | 40.96±3.44 | 48.24±2.41 | 49.60±0.45 | 30.69±0.50 | 49.11±1.46 | 36.13±1.54 | 5,370,880 |
| FedProx [20] | 39.62±2.36 | 46.74±1.24 | 49.56±0.50 | 31.40±0.50 | 52.16±1.75 | 36.13±1.54 | 5,370,880 |
| GCFL [46] | 56.98±5.56 | 59.68±3.37 | 48.93±1.02 | 30.84±0.36 | 49.44±2.95 | 34.87±0.68 | 10,741,760 |
| FedStar [38] | 54.76±1.28 | 56.49±0.86 | 51.89±0.33 | 36.89±0.43 | 58.28±0.53 | 39.97±1.22 | 416,000 |
| FGAD | **64.97±0.52** | **66.60±1.12** | **55.08±1.85** | **66.67±0.00** | **60.51±1.18** | **66.82±0.14** | 21,130 |

**Table 2: Anomaly detection performance (mean(%) ± std(%)) under the multi-dataset setting. Note that the best performance is marked in Bold.**

| Methods | MOLECULES | | BIOCHEM | | SOCIALNET | | MIX | |
|---|---|---|---|---|---|---|---|---|
| | AUC | AUPRC | AUC | AUPRC | AUC | AUPRC | AUC | AUPRC |
| Self-train | 61.26±2.91 | 61.31±1.91 | 54.54±0.99 | 52.29±0.40 | 50.31±1.55 | 39.96±1.58 | 51.94±0.42 | 47.65±0.64 |
| FedAvg [33] | 54.41±3.21 | 55.55±3.23 | 40.88±1.36 | 51.63±1.13 | 48.21±1.02 | 38.29±1.29 | 47.96±0.61 | 44.89±0.68 |
| FedProx [20] | 57.93±2.14 | 58.72±2.25 | 46.04±0.49 | 51.57±0.80 | 47.26±0.10 | 37.23±0.92 | 46.79±0.63 | 44.19±0.29 |
| GCFL [46] | 45.67±1.33 | 51.96±0.79 | 41.49±0.30 | 52.23±0.65 | 47.59±0.95 | 37.53±0.93 | 49.58±0.50 | 45.37±0.69 |
| FedStar [38] | 56.15±0.92 | 59.73±1.21 | 47.80±0.48 | 56.48±0.19 | 53.79±2.03 | 36.40±1.11 | 50.53±1.11 | 45.83±0.41 |
| FGAD | **62.15±0.69** | **79.19±0.49** | **58.09±0.85** | **59.04±0.54** | **54.86±0.29** | **56.88±0.98** | **58.14±0.36** | **52.03±0.63** |

(BIOCHEM) and mix data types (MIX). This allows us to thoroughly assess FL-based graph anomaly detection across a spectrum of data types and collaboration scenarios.

Note that for all datasets, we regard the graphs in the first class as normal and graphs in other classes as anomalous. We allocate 80% of the normal graphs data for training, and subsequently construct the testing data by combining the remaining normal data with an equal number of anomalous graphs. Refer to **Appendix 2** for more information and construction details of each dataset.

***Network Structure.*** We employ a 3-layer GIN [47] as the backbone network for our method, with the aggregated dimension in each layer set to 64. Additionally, we adopt the 4-layer and 3-layer fully connected networks for the teacher head and student head, respectively. The network structure of the teacher head is set to 256-192-128-64-2, while for the student head is 192-128-64-2. Refer to our code for further details and reproducibility[2].

***Baseline Methods.*** We compare the proposed FGAD method with several state-of-the-art baseline methods. We include two federated learning methods: FedAvg [33] and FedProx [20], as well as two federated graph learning methods: GCFL [46] and FedStar [38]. Note that in order to adapt these baseline methods to the graph anomaly detection task, we integrate them with DeepSVDD [37] to construct an end-to-end graph anomaly detection model. Besides, we regard the self-training strategy without the FL setting as one of

the baselines. To ensure a fair comparison with FGAD, we employ the same GIN network structure as FGAD in all baseline methods.

***Implementation Details.*** We use GIN [47] as the graph representation learning backbone for FGAD and all baselines. The number of GIN layer $K$ is set to 3, and the dimensions of the hidden layer of GIN and projection head of student and teacher models are all set to 64. We utilize Adam [13] as the optimizer and fixed the learning rate $\alpha = 0.001$. For all datasets, we first pretrain the anomaly generator and teacher model for 10 epochs, thereafter, jointly train with knowledge distillation and collaborative learning for 200 epochs. The implementation of FGAD is based on PyTorch Geometric [9] library, and the experiments are run on NVIDIA Tesla A100 GPU with AMD EPYC 7532 CPU.

***Evaluation Metrics:*** We use Area Under the Curve (AUC) and Area Under the Precision-Recall Curve (AUPRC) as the evaluation metrics in the experiment. Each method is executed 10 times to report their means and standard deviations.

Additionally, refer to **Appendix 3** for more experimental settings, including the training details, trade-off parameter setting, and baseline setting.

## 4.2 Experimental Results

In this section, we conduct comprehensive experiments including two types of non-IID graph scenarios, *i.e.*, the single-dataset and multi-dataset distributed in multiple clients, to validate the effectiveness of the proposed method. Table 1 and Table 2 show the

---

[2]Code is available at https://github.com/wownice333/FGAD.

experimental results of FGAD and several state-of-the-art baselines, from which we can have the following observations.

- **Comparison:** In the single-dataset experiment, FGAD demonstrates a remarkable advantage over all baseline methods. For instance, in the IMDB-BINARY dataset, FGAD achieves significant performance improvement, exceeding Self-train by 23.39% in AUC and 19.17% in AUPRC. It also significantly surpasses classical FedAvg and FedProx. Furthermore, FGAD outperforms the state-of-the-art baselines GCFL and FedStar by a substantial 7.99% and 10.21% in AUC, respectively. Similar trends are observed across other benchmarks, demonstrating the effectiveness of FGAD. In the multi-dataset experiment, the GAD task is more challenging as the non-IID problem is more severe than the single-dataset scenario. Nevertheless, FGAD still exhibits outstanding performance compared to other baselines. For example, on MOLECULES, FGAD outperforms the runner-up FedStar by 6.00% in AUC and 19.46% in AUPRC. Besides that, it achieves more than a 10.00% performance improvement compared to other baseline methods. More importantly, we can observe from Table 1 that FGAD significantly reduces communication costs during collaborative learning compared to other baseline methods.
- **Discussion:** The Self-train strategy discards collaborative training and fails to leverage the knowledge from other clients to learn more robust local GAD models. FedAvg and FedProx require the transmission of all network parameters of the local models, which introduces severe non-IID problems in collaborative learning. Consequently, these three baselines yield sub-optimal performance in most cases. Although GCFL incorporates a specific design to alleviate non-IID challenges, such as utilizing clustered FL for collaborative learning, it still necessitates the transmission of all network parameters and does not effectively address non-IID problems, as validated by the experimental results. On the other hand, FedStar achieves runner-up performance in most cases, which may primarily be attributed to the introduced structural embedding that helps to preserve the personalization of local models. Compared with the baseline methods, FGAD considers enhancing the detecting capability of local models in a self-boosted manner, and introduces an effective collaborative learning mechanism by leveraging knowledge distillation. This allows FGAD to learn more powerful local GAD models, mitigate the adverse effects of non-IID problems, and reduce communication costs among clients.

## 4.3 Embedding Visualization

We employ t-SNE [40] to visualize the learned embeddings for intuitive comparison. Figure 3 shows the embedding visualization for AIDS, one of the constituents of MOLECULES. We include results from FedAvg, GCFL, and FedStar for a comprehensive analysis. It's evident that the learned embeddings by FedAvg and GCFL exhibit poor discriminative properties, with both normal and anomalous graphs appearing entangled in the latent space. Although the visualization result of FedStar shows some separation between normal and anomalous graphs, decision boundaries remain blurred. Conversely, the learned embeddings of FGAD are clearly more discriminative compared to the other baseline methods. The visualization

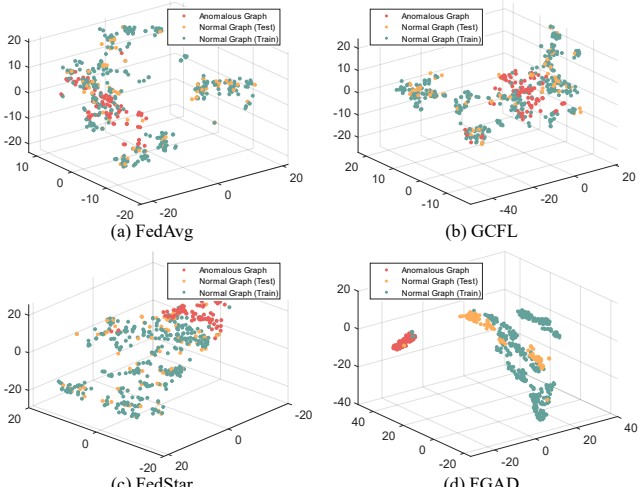

**Figure 3: Embedding visualization of the proposed FGAD compared with several baselines using t-SNE. Note that the data point marked in yellow, red, and green correspond to the normal graph (test), anomalous graph, and normal graph (train), respectively.**

**Table 3: Ablation study results (mean(%) ± std(%)) of FGAD and its three variants.**

| Methods | IMDB-MULTI | | MOLECULES | |
|---|---|---|---|---|
| | AUC | AUPRC | AUC | AUPRC |
| FGAD_v1 | 56.67±1.72 | 64.91±1.85 | 57.98±2.78 | 75.80±0.09 |
| FGAD_v2 | 56.69±1.22 | 65.98±0.90 | 59.41±2.22 | 77.32±1.02 |
| FGAD_v3 | 55.23±3.54 | 61.02±2.68 | 55.58±4.56 | 66.73±0.80 |
| FGAD | **60.51±1.18** | **66.82±0.14** | **62.15±0.69** | **79.19±0.49** |

of FGAD reveals distinct boundaries between the embeddings of normal and anomalous graphs, supporting its effectiveness.

## 4.4 Ablation Study

To validate the effectiveness of each component in the proposed FGAD method, we derive three variants from FGAD and perform a systematic evaluation. Specifically, we illustrate the construction details of the three variants as follows:

- **FGAD_v1:** This variant only considers local training in each client and abandons the collaborative learning between clients.
- **FGAD_v2:** This variant drops the proposed collaborative learning mechanism and follows the parameter aggregation mechanism of the classical FedAvg method.
- **FGAD_v3:** This variant drops the knowledge distillation module, *i.e.*, removes the student model and only takes the teacher anomaly detector in collaboration.

Table 3 shows the experimental results of FGAD and its three variants on two datasets, yielding the following observations. FGAD_v1 demonstrates a performance decline compared to FGAD, which is primarily due to the fact that FGAD_v1 exclusively focuses on

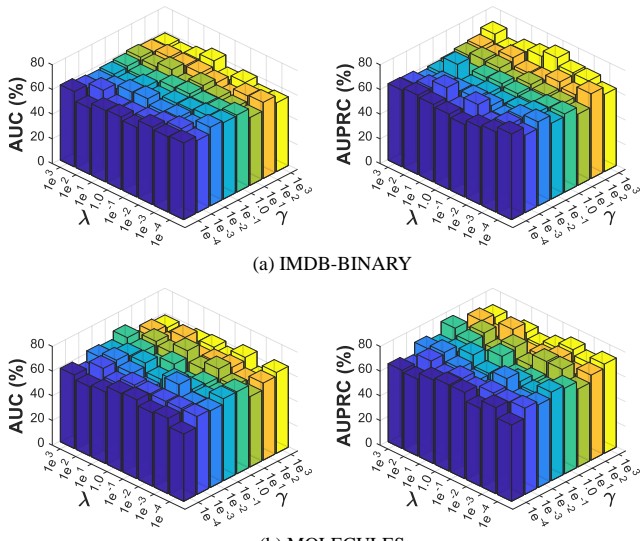

(a) IMDB-BINARY

(b) MOLECULES

**Figure 4: Parameter analysis of $\lambda$ and $\gamma$ on IMDB-BINARY and MOLECULES. Note that the values of $\lambda$ and $\gamma$ range from $[1e^{-4}, \ldots, 1e^3]$.**

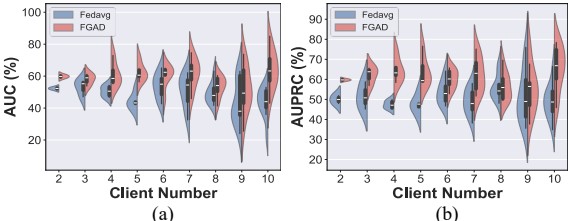

**Figure 5: Average performance and distribution of variance between clients of FedAvg and FGAD. Note that the client number is set to $[2, \ldots, 10]$.**

local training, neglecting collaboration with other clients. Consequently, it fails to leverage the comprehensive knowledge of other clients. Secondly, when we substitute the proposed collaborative learning mechanism with the classical FedAvg, there is also a noticeable performance decline. This can be attributed to the potential susceptibility of parameter transmission in FedAvg to the adverse effects of non-IID problems. Third, FGAD consistently outperforms FGAD_v3 by a significant margin. This observation reveals the crucial role of the self-boosted distillation module in maintaining the personalization of local models within each client, which effectively mitigates the non-IID problems. Overall, the ablation study results fully support the rationale and the effectiveness of each component proposed in FGAD.

### 4.5 Parameter Analysis

*4.5.1* ***Impact of Hyper-Parameters $\lambda$ and $\gamma$.*** The objective function of the proposed FGAD method contains two main hyper-parameters, *i.e.*, $\lambda$ and $\gamma$. In this section, we analyze the impact of the two hyper-parameters on anomaly detection performance. Specifically, we vary the values of $\lambda$ and $\gamma$ within the range of $[1e^{-3}, \ldots, 1e^4]$ and Figure 4 presents the experimental results on IMDB-BINARY and MOLECULES datasets. Based on the observations in the figure, we draw several conclusions. Firstly, FGAD tends to yield sub-optimal performance when the values of $\lambda$ and $\gamma$ are set too low, *e.g.*, $1e^{-4}$ and $1e^{-3}$. This emphasizes the significant role of both loss terms in the FGAD framework and suggests their effectiveness. Secondly, we can observe that excessively high values of $\lambda$ and $\gamma$ also have an adverse impact on performance, because they may obscure the primary objective of optimizing the anomaly detector. Finally, it is worth noting that FGAD exhibits relatively stable performance both in AUC and AUPRC across a wide range of $\lambda$ and $\gamma$ values, demonstrating its robustness.

*4.5.2* ***Impact of Client Numbers.*** The number of clients $C$ is another hyper-parameter in the FGAD framework, and its impact on the performance is crucial for assessing the scalability of client numbers. Therefore, we vary the number of clients $C$ within the range of $[2, \ldots, 10]$ and conduct the experiment. The results on IMDB-BINARY are reported in Figure 5. Note that we also include FedAvg as a baseline method for comparative analysis. It can be observed that FGAD consistently achieves remarkable performance improvement compared to FedAvg in all cases, and exhibits stability against changes in the number of clients. However, when the number of clients increases to certain large values, the average performance degrades, and the performance variance between different clients becomes more significant in both FGAD and FedAvg. This is primarily due to the gradually increasing discrepancy between the graph data distributed across different clients, which causes more severe non-IID problems. Nevertheless, FGAD still exhibits relatively smaller performance fluctuations compared with FedAvg, which fully demonstrates the scalability of the proposed FGAD method.

*4.5.3* ***Other Experiments.*** We further analyze the impact of the number of layers and latent dimensions in GIN, as well as the justification of backbone sharing. Refer to **Appendix 6**, **7**, and **8** for details due to the limitation of the paper length.

## 5 Conclusion

In this paper, we studied a challenging GAD problem with non-IID graph data distributed across multiple clients, and proposed an effective federated graph anomaly detection (FGAD) method to tackle this issue. To enhance the detecting capability of local models, we proposed to unsupervised train a classifier in a self-boosted manner by distinguishing the normal and anomalous graphs generated from an anomaly generator. Besides that, in order to alleviate the adverse impact of non-IID problems among clients, we introduce a student model to distill knowledge from the teacher anomaly detector and engage only the student model in collaborative learning so that the personalization of local models could be preserved. Furthermore, we improved the collaborative learning mechanism that streamlines the capacity of local models and reduces the communication costs during collaborative learning. Comparative experiments with state-of-the-art baselines in the graph anomaly detection tasks on single/multi datasets (from diverse domains) demonstrated the superiority of FGAD. We believe that this work will pave the way for future research on collaborative GAD under the FL setting.

## Acknowledgement

This research is supported by the National Research Foundation Singapore and DSO National Laboratories under the AI Singapore Programme (AISG Award No: AISG2-RP-2020-018). Any opinions, findings and conclusions or recommendations expressed in this material are those of the authors and do not reflect the views of National Research Foundation, Singapore. Additionally, this research is supported in part by the National Natural Science Foundation of China under Grant U21A20472.

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
