# OpenReview forum: "Towards Effective Federated Graph Anomaly Detection via Self-boosted Knowledge Distillation"
_acmmm.org/ACMMM/2024/Conference — MM2024 Poster_

### Official Review · Reviewer_sgXq · 2024-05-23

**Rating:** 3
**Confidence:** 3

**Summary:**

This paper studies a valuable topic in graph anomaly detection. The authors propose an effective federated graph anomaly detection framework. They design a student model to distill the knowledge from the trained teacher anomaly detector. Then they align the student model’s predicted distributions with that of the teacher model. Moreover, the authors further design an effective collaborative learning mechanism and achieve good performance.

**Strengths:**

This paper investigates the challenging GAD issue on non-IID graphs under federated learning setup, and propose an effective federated graph anomaly detection (FGAD) method.

**Limitations:**

1. What is non-IID property in federated graph anomaly detection? The authors should provide more explanations in Introduction Section and clearly articulate the advantages of the model in solving the problem of non-IID.
2. The authors should provide more explanations of Eq.(8), e.g., A_c^{ij}.
3. The anomalous graphs in training are generated by perturbing graph structure. Why not use other methods? e.g. perturbing node features. Does this fit the real-life scenario?
4. The compared methods in experiments are old. Why doesn't the author compare the proposed method with [1]? The experiment results cannot confirm the effectiveness of the proposed model.

[1] Ma R, Pang G, Chen L, et al. Deep graph-level anomaly detection by glocal knowledge distillation[C]//Proceedings of the fifteenth ACM international conference on web search and data mining. 2022: 704-714.

**Suitability:**

3

---

### Official Review · Reviewer_URpu · 2024-05-24

**Rating:** 2
**Confidence:** 3

**Summary:**

In this paper, the authors propose a federated graph anomaly detection (FGAD) framework to address the challenges posed by the heterogeneity and non-IID property of graph data in graph federated learning. The FGAD framework utilizes an anomaly generator to create anomalous graphs and unsupervised trains a local teacher model. Then a student model is used to distill knowledge from the trained teacher model, ensuring that its prediction distribution of normal graphs aligns with that of the teacher model, thereby mitigating the impact of non-IID problems. Finally, a collaborative learning mechanism is employed to globally optimize the parameters of the student head.

**Strengths:**

1. The completeness of the method proposed in this paper is well, as it is the first to tackle graph anomaly detection on non-IID graphs with a federated learning setup and introduces a self-boosted distillation module to preserve the model personalization and alleviate non-IID problems.
2. This paper completed comparative experiments of multiple baseline methods in single-dataset and multi-data scenarios, as well as some ablation experiments of its own.

**Limitations:**

1. The backbone used in both the teacher and student models is solely GIN. There is a lack of experiments using other GNNs to demonstrate the generalizability of the method.
2. The experimental setup for multi-dataset scenario is unclear. The paper doesn’t specify how the multi-domain datasets are distributed to multiple clients.
3. The paper mentions that all normal graphs are simply regarded as one class, while the generated anomalous graphs are regarded as abnormal. Under this setting, is it possible that a perturbed graph from one class might be considered a normal graph in another class?
4. There is no clear explanation in the experimental section about how the abnormal data is obtained.

**Suitability:**

1

---

### Official Review · Reviewer_kQNu · 2024-05-24

**Rating:** 4
**Confidence:** 3

**Summary:**

This paper proposes an effective federated graph anomaly detection framework (FGAD). It conducts federated learning across multiple clients, not sharing local data, thereby preventing leakage of sensitive information and protecting client privacy.

**Strengths:**

1.The viewpoints presented in this paper are highly innovative.
2.The experiments conducted are comprehensive and thorough.

**Limitations:**

1.Why only upload the parameter set of the student head for collaboration instead of uploading all the network parameters? Even though the student models in each client share the parameters of the GNN portion with the teacher model, the parameters across different clients are still different. Similarly, w_g is different for different clients, so why not upload?
2. Personally, I believe that the ablation section is not sufficiently comprehensive. Ablation experiments for the anomaly generator should also be included.
3. A minor detail: should "a node set and edge set" be adjusted to "a node set and an edge set" or "node set and edge set"?
4. After reading this paper, the feeling is that although the paper is innovative, its relationship with multimodal and multimedia aspects is not significant.

**Suitability:**

1

---

### Official Review · Reviewer_64fN · 2024-05-30

**Rating:** 4
**Confidence:** 2

**Summary:**

This paper proposes a federated graph anomaly detection (FGAD) framework to enable collaborative learning among clients while preserving data privacy. The key components of FGAD include an anomaly generator to produce diverse anomalous graphs, a self-boosted distillation module to maintain the personalization of local models, and an effective collaborative learning mechanism to reduce communication costs. Experiments on various datasets demonstrate the superiority of FGAD over state-of-the-art baselines in detecting anomalies in non-IID graph data.

**Strengths:**

Pros:
1. FGAD provides a privacy-preserving solution for collaborative graph anomaly detection by leveraging federated learning.
2. The self-boosted distillation module enhances the detecting capability of local models by generating diverse anomalous graphs and preserving personalization.
3. The collaborative learning mechanism streamlines the capacity of local models and significantly reduces communication costs.
4. FGAD demonstrates superior performance compared to state-of-the-art baselines on both single-dataset and multi-dataset scenarios.
5. Extensive experiments, including ablation studies and parameter analysis, validate the effectiveness and robustness of FGAD.

**Limitations:**

Cons:
1. Statistical testing is desired. It is highly recommended that the authors conduct a Statistical testing. Otherwise, the statistical significance of the results are not clear.
2. No code is provided. It is strongly encouraged that the authors release the code for reproduction. The authors could upload code to website such as https://anonymous.4open.science/

**Suitability:**

2

---

### Meta-Review · Area_Chair_5671 · 2024-07-05

**Recommendation:** Accept (Poster)
**Confidence:** 4

**Metareview:**

The authors have proposed a federated graph anomaly detection framework that uses an anomaly generation module to perturb the normal graphs and then trains an anomaly detector by distinguishing between generated anomalous and normal graphs. The framework also uses a student model to distill knowledge from the trained anomaly detector, thus reducing the impact of non-IID data. Finally, it contains also a collaborative learning mechanism that has the goal of preserving the personalization of local models. While some reviewers are keeping some concerns about the paper, the authors were able in the original submission and also in the rebuttal to show the value of the proposed framework, for example adding further comparisons with SOTA approaches and reporting statistical significance of the results by reporting a t-test in the rebuttal (this should be included in the potential final version). Again, the ablation reported seems comprehensive.